# VISUAL EXPLANATION FOR DEEP METRIC LEARNING

## ABSTRACT

This work explores the visual explanation for deep metric learning and its applications. As an important problem for learning representation, metric learning has attracted much attention recently, while the interpretation of such model is not as well studied as classification. To this end, we propose an intuitive idea to show where contributes the most to the overall similarity of two input images by decomposing the final activation. Instead of only providing the overall activation map of each image, we propose to generate point-to-point activation intensity between two images so that the relationship between different regions is uncovered. We show that the proposed framework can be directly deployed to a large range of metric learning applications and provides valuable information for understanding the model. Furthermore, our experiments show its effectiveness on two potential applications, i.e. cross-view pattern discovery and interactive retrieval.

## 1 INTRODUCTION

Learning the similarity metrics between arbitrary images is a fundamental problem for a variety of tasks, such as image retrieval (Oh Song et al. (2016)), verification (Schroff et al. (2015); Luo et al. (2019)), localization (Hu et al. (2018)), video tracking (Bertinetto et al. (2016)), etc. Recently the deep Siamese network (Chopra et al. (2005)) based framework has become a standard architecture for metric learning and achieves exciting results on a wide range of applications (Wang et al. (2019)). However, there are surprisingly few papers conducting visual analyses to explain why the learned similarity of a given image pair is high or low. *Specifically, which part contributes the most to the similarity is a straightforward question and the answer can reveal important hidden information about the model as well as the data.*

Previous visual explanation works mainly focus on the interpretation of deep neural network for classification (Springenberg et al. (2014); Zhou et al. (2016; 2018); Fong & Vedaldi (2017)). Guided back propagation (Guided BP) (Springenberg et al. (2014)) has been used for explanation by generating the gradient from prediction to input, which shows how much the output will change with a little change in each dimension of the input. Another representative visual explanation, class activation map (CAM) Zhou et al. (2016), generates the heatmap of discriminative regions corresponding to a specific class based on the linearity of global average pooling (GAP) and fully connected (FC) layer. However, the original method only works on this specific architecture configuration and needs retraining for visualizing other applications. Based on the gradient of the last convolutional layer instead of the input, Grad-CAM (Selvaraju et al. (2017)) is proposed to generate activation maps for all convolutional neural network (CNN) architectures. Besides, other existing methods explore network ablation (Zhou et al. (2018)), the winner-take-all strategy (Zhang et al. (2018)), inversion (Mahendran & Vedaldi (2015)), and perturbation (Fong & Vedaldi (2017)) for visual explanation.

Since the verification applications like person re-identification (re-ID) (Luo et al. (2019)) usually train metric learning models along with classification, recent work (Yang et al. (2019)) starts to leverage the classification activation map to help improve the overall performance, but the activation map of metric learning is still not well explored. For two given images, a variant of Grad-CAM has been used for visualization of image retrieval (Gordo & Larlus (2017)) by computing the gradient from the cosine similarity of the embedding features to the last convolutional layers of both images. However, Grad-CAM only provides the overall highlighted regions of two input images, the relationship between each activated region of two images is yet to be uncovered. Since the similarity is calculated from two images and possibly based on several similar patterns between them, the relationship between these patterns is critical for understanding the model.

In this paper, we propose an activation decomposition framework for visual explanation of deep metric learning and explore the relationship between each activated region by point-to-point activation response between two images. As shown in Fig. 1, the overall activation map of the proposed method is generated by decomposing the similarity along each image. In this example, the query image (A) has a high activation on both the eyes and mouth areas, but the overall map of the retrieved image (B) only highlights the eyes. It is actually hard to understand how the model works only based on the overall maps. For image B, the mouth region (green point) has a low activation which means the activation between the mouth and **the whole image A** is low compared to the overall activation (similarity). However, by further decom-

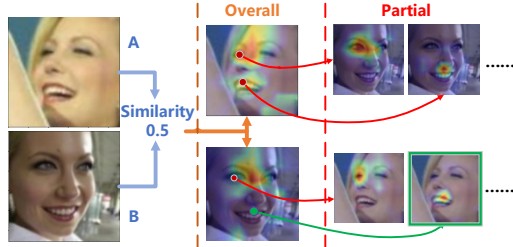

Figure 1: An overview of activation decomposition. The overall activation map on each image highlights the regions contributing the most to the similarity. The partial activation map highlights the regions in one image that have large activation responses on a specific position in the other image, e.g. mouth or eye.

posing this activation (green point) along image A, that is to give the activation between the mouth region of image B and **each position in image A**, the resulting partial (or point-specific) activation map (green box) reveals that the mouth region of image B still has a high response on the mouth region of image A. This partial activation map can be generated for each pixel, which renders the point-to-point activation intensity representing the relationship between regions in both images, e.g. eye-to-nose or mouth-to-mouth. The partial activation map provides much more refined information about the model which is crucial for explanation. The contributions are summarized as follows.

- We propose a novel explanation framework for deep metric learning architectures and it may serve as an analysis tool for a host of applications, e.g. face recognition, person re-ID.

- The proposed method uncovers the point-to-point activation intensity which is not explored by existing methods. Our experiments further show the importance of the partial activation map on several new applications, i.e. cross-view pattern discovery and interactive retrieval.

- Our analysis suggests that two widely believed arguments (Section 2.1, 4) about CAM and Grad-CAM are inaccurate.

## 2 RELATED WORK

### 2.1 INTERPRETATION FOR CLASSIFICATION

As a default setting, most existing interpretation methods (Zhou et al. (2016); Springenberg et al. (2014); Zhang et al. (2018); Fong & Vedaldi (2017)) are designed in classification context where the output score is generated from one input image. Among the approaches that do not require architecture change, one intuitive idea (Zhou et al. (2016); Zhang et al. (2018)) is to look into the model and see where contributes the most to the final prediction. CAM and its variant Grad-CAM are among the most widely used approaches, but recent works simply consider CAM as a heuristic linear combination of convolutional feature maps which is limited to the original architecture. It is also believed (Zhou et al. (2016; 2018); Selvaraju et al. (2017)) that CAM only applies to the specific architecture configuration of GAP and one FC layer (Argument 1). On the contrary, we argue that CAM is only a special case of activation decomposition on the specific architecture, and the idea applies to much more architectures, even beyond the classification problem. It is also believed that Grad-CAM is a generalization of CAM for arbitrary CNN architecture (Argument 2), but the original paper only provides the proof on CAM's architecture. We show that Grad-CAM is not equivalent to activation decomposition for some architectures (Section 4.3). Another direction of explanation is to check the black box model by modifying the input and observing the response in the prediction. A representative method Fong & Vedaldi (2017) aims to optimize the blurred region and see which region has the strongest response on the output signal when it is blurred. Perturbation optimization is applicable for any black box model, but the optimization can be computational expensive.

## 2.2 INTERPRETATION FOR METRIC LEARNING

Guided BP (Springenberg et al. (2014)) can be easily adopted to metric learning, but recent works (Adebayo et al. (2018); Fong & Vedaldi (2017)) claim that gradient can be irrelevant to the model and guided-BP fails on the sanity check (Adebayo et al. (2018)). Among methods that pass the sanity check, Grad-CAM has been used for visualization of image retrieval (Gordo & Larlus (2017)), but no quantitative result is provided. Instead of adding more gradient heuristics like Grad-CAM++ (Chattopadhay et al. (2018)), we propose to explain the result of Grad-CAM under the decomposition framework. As shown in Section 4.3, by removing the heuristic step in Grad-CAM, the meaning of the generated activation map can be explained as the first two terms of the proposed framework. The perturbation optimization needs reformulation for metric learning and can be more computational expensive if the point-to-point activation map is desired.

## 3 ACTIVATION DECOMPOSITION ON A SIMPLE ARCHITECTURE

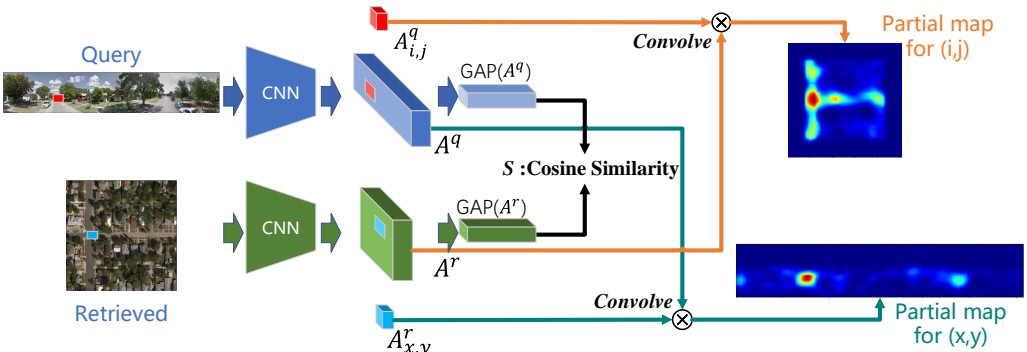

Figure 2: Activation decomposition on a simple architecture for deep metric learning.

To better introduce our method, we first review the formulation of CAM and illustrate our idea on a simple architecture (CNN+GAP as in Fig. 2) for metric learning. As shown in Eq. 1, CAM is actually a spatial decomposition of the prediction score of each class, and the original method only applies for CNN with GAP and one FC layer without bias. The decomposition clearly shows how much each part of the input image contributes to the overall prediction of one class and provides valuable information about how the decision is made inside the classification model. The idea is based on the linearity of GAP:

$$S_c = \sum_k \omega_{k,c}(\frac{1}{Z}\sum_{i,j} A_{i,j,k}) = \frac{1}{Z}\sum_{i,j}(\sum_k \omega_{k,c}A_{i,j,k}). \tag{1}$$

Here $S_c$ denotes the overall score (before softmax) of class $c$ and $\omega_{k,c}$ is the FC layer parameter for the $k$-th channel of class $c$. $A_{i,j,k}$ denotes the feature map of the last convolutional layer at position $(i,j)$, and $Z$ is the normalization term of GAP. In fact, the result $\sum_k \omega_{k,c}A_{i,j,k}$ can be considered as a decomposition of $S_c$ along $(i,j)$. For a two-stream architecture (see Fig. 2) with GAP and the cosine similarity metric ($S$) for metric learning application (e.g. image retrieval), we propose to do decomposition along $(i,j,x,y)$ so that the relationship between different parts of two images are uncovered:

$$S = \sum_k GAP(A_k^q)GAP(A_k^r) = \frac{1}{Z}\sum_k(\sum_{i,j} A_{i,j,k}^q \sum_{x,y} A_{x,y,k}^r) = \frac{1}{Z}\sum_{i,j,x,y}(\sum_k A_{i,j,k}^q A_{x,y,k}^r). \tag{2}$$

Here the normalization terms of the cosine similarity (L2 norm) and GAP are included in $Z$. We use $(i,j)$, $(x,y)$ for different streams because cross-view applications (Hu et al. (2018)) may have different image sizes for two streams. $A$ denotes the feature map of the last convolutional layer. The superscripts $q$ and $r$ respectively denote the query and the retrieved image in this paper. For each query point $(i,j)$ in the query image, the corresponding activation map in the retrieved image is given by $\sum_k A_{i,j,k}^q A_{x,y,k}^r$ which is the contribution of features at $(i,j,x,y)$ to the overall cosine similarity. Like CAM and Grad-CAM, bilinear interpolation is implemented to generate the contribution of each pixel pair and the full resolution map. The overall activation maps of two images

are generated by a simple summation along $(i, j)$ or $(x, y)$. Although we only show the positive activation in the map, the negative value is still available for counterfactual explanation (Selvaraju et al. (2017)).

Since recent works (Liu et al. (2017); Wang et al. (2018)) have highlighted the superiority of L2 normalization, we use the cosine similarity $S$ as the default metric. With L2 normalization, the squared Euclidean distance $D$ equals to $2 - 2S$ so that $S$ and $D$ are equivalent as metrics. Although there are still a number of existing works (Luo et al. (2019); Ristani & Tomasi (2018)) training with the Euclidean distance without L2 normalization, Luo et al. (2019) shows that the cosine similarity still performs well as the evaluation metric. We further empirically show that the cosine similarity works well for this case (Section 5.4).

## 4 EXTENSION FOR COMPLEX ARCHITECTURES

Recent metric learning approaches usually leverage more complex architectures to improve performance, e.g. adding several FC layers after the flattened feature or global pooling. Although different metric learning applications have different head architectures (e.g. GAP+FC) on CNN, the basic components are highly similar. We introduce a unified extension to make our method applicable to most existing state-of-the-art architectures for various applications, including image retrieval (Wang et al. (2019)), face recognition (Deng et al. (2019)), re-ID (Luo et al. (2019)) and geo-localization (Hu et al. (2018)). *Note that the extension also applies to classification architectures.* In Section 4.1, we address linear components by considering them together as one linear transformation. Then Section 4.2 focuses on transforming nonlinear components to linear in the validation phase.

### 4.1 LINEAR COMPONENT

The GAP can be considered as a special case of flattened layer (directly reshape the last convolutional layer $A \in \mathbb{R}^{m \times n \times p}$ as one vector $\hat{A} \in \mathbb{R}^{mnp}$ without pooling) multiplied by a transformation matrix $T_{GAP} \in \mathbb{R}^{l \times mnp}$ where $l$ denotes the length of the feature embedding vector (see Appendix A.3 for details). Without loss of generality, we consider a two-stream framework with flattened layer followed by one FC layer with weights $\hat{W} \in \mathbb{R}^{l \times mnp}$ and biases $B \in \mathbb{R}^l$. Since the visual explanation is generated at the test phase when typical components such as FC layer and batch normalization (BN) are linear, all the linear components together are formulated as one linear transformation $g(\hat{A}) = \hat{W}\hat{A} + B = \sum_{i,j} W_{i,j} A_{i,j} + B$ in the FC layer. Here $W_{i,j} \in \mathbb{R}^{l \times p}$ and $A_{i,j} \in \mathbb{R}^p$ denote the weights matrix and feature vector corresponding to position $(i, j)$. Although $B$ is ignored in CAM, we keep it as a residual term in the decomposition. Then Eq. 2 is re-formulated as:

$$
\begin{aligned}
SZ &= g^q(A^q) \cdot g^r(A^r) \\
&= \left(\sum_{i,j} W_{i,j}^q A_{i,j}^q + B^q\right) \cdot \left(\sum_{x,y} W_{x,y}^r A_{x,y}^r + B^r\right) \\
&= \sum_{i,j,x,y} (W_{i,j}^q A_{i,j}^q) \cdot (W_{x,y}^r A_{x,y}^r) + \sum_{i,j} (W_{i,j}^q A_{i,j}^q \cdot B^r) + \sum_{x,y} (W_{x,y}^r A_{x,y}^r \cdot B^q) + B^q \cdot B^r.
\end{aligned}
\tag{3}
$$

Here $Z$ denotes the normalization term for cosine similarity (L2 norm), and $\cdot$ is the inner product. The decomposition of $S$ has four terms, and the last three terms contain the bias $B$. The first term clearly shows the activation response for location pair $(i, j, x, y)$ and is considered as the **point-to-point activation** in this paper. The second and third terms correspond to the activation between one image and the bias of the other image. Although they can be considered as negligible bias term, they actually contribute to the overall activation map. For the overall activation map of the query image, the second term can be included, because $W_{i,j}^q A_{i,j}^q \cdot B^r$ varies at different $(i, j)$ positions while the third and last terms stay unchanged. Similarly, the first and third terms are considered when calculating the overall map for the retrieved image. We investigate both settings about the bias term (i.e. w or w/o) on the overall activation map and they are referred to as "Decomposition": $(W_{i,j}^q A_{i,j}^q) \cdot (\sum_{x,y} W_{x,y}^r A_{x,y}^r)$, and "Decomposition+Bias" : $(W_{i,j}^q A_{i,j}^q) \cdot (\sum_{x,y} W_{x,y}^r A_{x,y}^r + B^r)$ in Section 5. The last term is pure bias which is the same for every input image. Therefore, we ignore this term if not mentioned.

### 4.2 NON-LINEAR COMPONENT

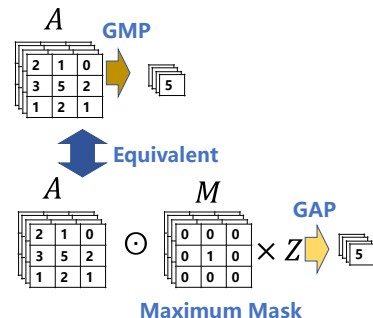

Non-linear transformation increases the difficulty of activation decomposition, fortunately, the most widely used non-linear components – global maximum pooling (GMP) and rectified linear unit (ReLU) – can be transformed as linear operations **in the validation phase** by multiplying a mask[1]. We only focus on GMP and ReLU in this paper, and all the mentioned components together have covered all the popular architectures for deep metric learning applications. In the validation phase, the GMP can be considered as a combination of a maximum mask $M$ and GAP or flattened layer as shown in Fig. 3. The result of Hadamard product ($M \odot A$) is considered as the new feature map which can be directly applied to Eq. 3. Similarly, by adding a mask for ReLU, the FC layer with ReLU can be included in $W$ and $B$ in the validation phase.

Figure 3: Maximum mask for GMP. $Z$ is the constant normalization term for GAP.

### 4.3 RELATIONSHIP WITH GRAD-CAM

Grad-CAM is another way for generating overall activation map on more complex architecture. When we calculate the activation map based on Grad-CAM, what do we get? Selvaraju et al. (2017) has shown that Grad-CAM is equivalent to CAM on GAP based architecture. However, for a classification architecture with flattened feature and one FC layer, the prediction score for class $c$ may be formulated as $S_c = \sum_{i,j,k} W_{i,j,k,c} A_{i,j,k}$ where $W$ is the reshaped weights of FC layer following the shape of the last convolutional feature ($A$). From our activation decomposition perspective, the contribution of each position $(i, j)$ is clearly given by $\sum_k W_{i,j,k,c} A_{i,j,k}$, while the Grad-CAM map for class $c$ at $(i, j)$ position is given by:

$$GradCAM_{i,j,c} = \sum_k (A_{i,j,k} GAP(\frac{\partial S_c}{\partial A_k})) = \sum_k (A_{i,j,k} GAP(W_k)) \tag{4}$$

In this case, the result of Grad-CAM is different from activation decomposition because of the GAP in Eq. 4. For architectures using GMP, they are also different, because only the maximal value of each channel contributes to the overall prediction score, while Grad-CAM puts the same weight for features at all positions (see Appendix A.4 for details). Although Selvaraju et al. (2017) empirically shows that the heuristic GAP step can help improve the performance of localization, this step makes the meaning of the generated map unclear. By removing this step, which means combining the gradient and feature directly as $A_{i,j,k} \frac{\partial S_c}{\partial A_{i,j,k}}$, the modified version would generate the same result as activation decomposition for flattened and GMP based architectures. As for the metric learning architecture discussed in Section 4.1, the Grad-CAM map of query image is written as (the derivation is included in Appendix A.5):

$$GradCAM_{i,j} = \frac{1}{Z} (\frac{\partial(E^q/|E^q|)}{\partial E^q} GAP(W^q) A^q_{i,j}) \cdot (\sum_{x,y} W^r_{x,y} A^r_{x,y} + B^r) \tag{5}$$

$E^q$ denotes the embedding vector of query image before L2 normalization and $Z$ is the normalization term. The gradient term $\frac{\partial(E^q/|E^q|)}{\partial E^q}$, which comes from the L2 normalization, would put less weights on dominant channels so that the generated activation map becomes more scattering (Appendix A.5). It can be removed by calculating gradient from $E^q \cdot E^r$ without L2 normalization. If we remove the gradient term as well as the GAP term, the result is actually equivalent to the overall map of "Decomposition+Bias" given by the first two terms of Eq. 3 as $(W^q_{i,j} A^q_{i,j}) \cdot (\sum_{x,y} W^r_{x,y} A^r_{x,y} + B^r)$.

## 5 EXPERIMENT

### 5.1 WEAKLY SUPERVISED LOCALIZATION

Weakly supervised localization has been adopted as evaluation metric in most existing works, we thus conduct localization experiment in the context of metric learning to evaluate the overall ac-

---

[1]The mask is computed once for each input image.

tivation map. We follow the state-of-the-art image retrieval method Wang et al. (2019) on CUB (Welinder et al. (2010)), which contains over 10,000 images from 200 bird spices. CUB is a challenging dataset for metric learning and the bounding box is available for localization evaluation. We first train a model on CUB with the loss function and architecture of Wang et al. (2019), and then conduct the weakly supervised localization to show the differences between the variants of proposed methods. As CAM and Grad-CAM generate the mask with different threshold settings (0.2 and 0.15), we use multiple thresholds and find the result is sensitive to threshold (see Appendix A.1). We report the best result of each method in Table1. The "Grad-CAM (no norm)" means the Grad-CAM with gradient computed from the product of two embedding vectors without normalization and the "Decomposition+Bias" denotes the first two terms of Eq. 3. The proposed framework outperforms Grad-CAM for metric learning as well as CAM based on classification. As justified in Section 4.3, computing the gradients before normalization does improve the performance of Grad-CAM by a large margin. Since the architecture of Wang et al. (2019) is based on GMP, "Grad-CAM (no norm)" is not equivalent to "Decomposition+Bias" as shown in Section 4.3.

Table 1: Localization accuracy for 0.5 IOU on CUB validation set.

| | Method | Accuracy |
|---|---|---|
| Classification | CAM (Zhou et al. (2016)) | 41.0% |
| | Grad-CAM | 16.7% |
| Metric Learning | Grad-CAM (no norm) | 48.3% |
| | Decomposition+Bias (ours) | 44.9% |
| | Decomposition (ours) | **50.6%** |

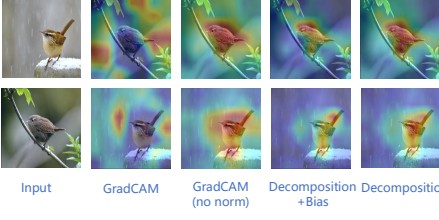

Figure 4: Qualitative results on CUB.

## 5.2 MODEL DIAGNOSIS

A large number of loss functions have been proposed for metric learning (Schroff et al. (2015); Oh Song et al. (2016); Wang et al. (2019)), while only the performance and embedding distribution are evaluated. Here we show that the overall activation map can also help evaluate the generalization ability of different metric learning methods. We follow the setting of Wang et al. (2019) and train

Table 2: Top-1 recall accuracy on CUB. 

Table 3: Localization accuracy (0.5 IOU) on CUB training set.

| Loss | Train | Validation |
|---|---|---|
| MS | 82.69% | 65.45% |
| Triplet | 82.98% | 60.55% |

| | Decomposition+Bias | Decomposition |
|---|---|---|
| MS | 52.10% | 58.99% |
| Triplet | 39.93% | 55.60% |
| Accuracy Drop | 12.17% | 3.39% |

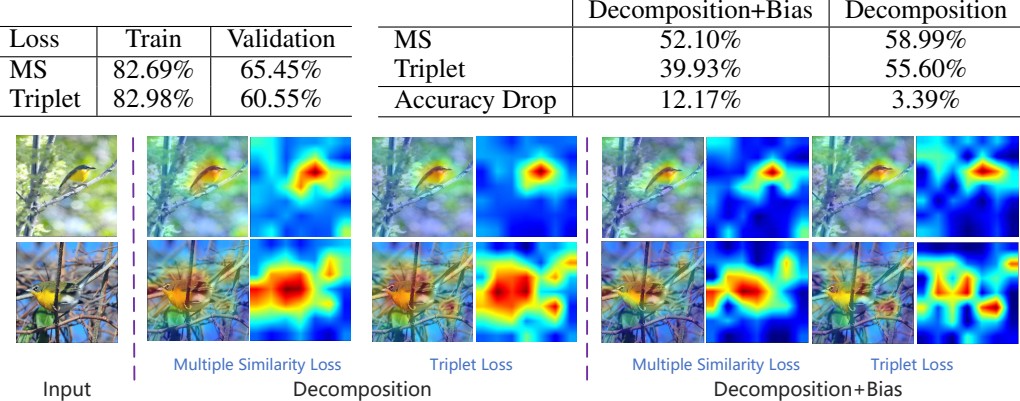

Figure 5: Qualitative results for model diagnosis.

two metric learning models with multiple similarity (MS) loss Wang et al. (2019) and triplet loss Schroff et al. (2015), respectively. As shown in Table 2, although they have almost the same accuracy on training set, there is a big gap between their generalization ability on the validation set. Before checking the result on the validation set, is there any clue in the training set for such gap? Despite the similar training accuracies, the predictions of two models are actually based on different regions (see Fig. 5). The quantitative results in Table 3 also support the fact that "Triplet" model is more likely to focus on the background region rather than the object as we witness the localization accuracy drop

for Triplet loss. Although "Decomposition+Bias" can be more sensitive to different loss functions, implying that the bias term does provide valuable information on overall map, both settings of the proposed method have the same trend on accuracy. Therefore, our activation decomposition framework can help shed light on the generalization ability of loss functions for metric learning.

## 5.3 APPLICATION I: CROSS-VIEW PATTERN DISCOVERY

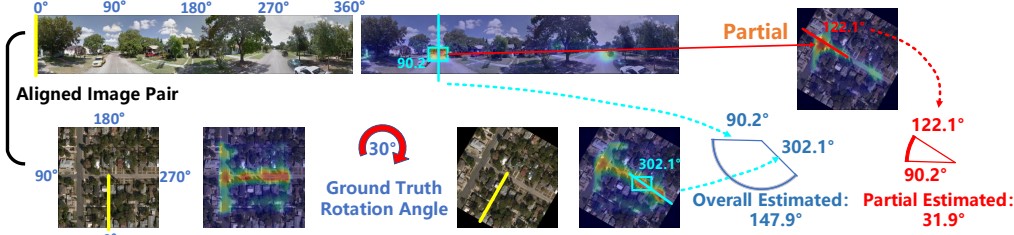

Figure 6: Example of cross-view pattern discovery, e.g. image orientation estimation. (Best viewed in color)

When the metric learning is applied to cross-view applications, e.g. image retrieval (Zhai et al. (2017); Hu et al. (2018); Tian et al. (2017)), the model is capable of learning similar patterns in different views which may provide geometric information of the two views, e.g. the camera pose or orientation information. We conduct orientation estimation experiment to show the advantage of partial activation map compared with the overall activation map on providing geometric information based on cross-view patterns. In our experiment, we take street-to-aerial image geo-localization as an example and train a Siamese network following the loss function of Hu et al. (2018). The objective of geo-localization is to find the best matched *aerial-view image* in a reference dataset for a query *street-view* image. We conduct the experiment on CVUSA (Zhai et al. (2017)), which is the most popular benchmark for this problem containing 35,532 training pairs and 8,884 test pairs. Each pair consists of a query street image and the corresponding orientation-aligned aerial image at the same GPS location. The orientation example is illustrated as angle degree in Fig. 6 (the yellow lines on both images correspond to 0°). We train the Siamese image matching model with randomly rotated aerial images so that the activation map is approximately rotation invariant for aerial image. As shown in Fig. 6, the overall activation map may contain multiple highlighted regions contributing to the overall matching/similarity score. Our method can further provide detailed point-to-point relationship which is critical for orientation estimation. Here we utilize the maximum activated region for orientation estimation to show the superiority of the partial map compared with the overall map. We first generate the overall activation map as well as the partial map corresponding to the maximum activation position of the query street-view image. For both activation maps, the activation peaks from different views are selected for orientation estimation as shown in Fig. 6.

In this example, the overall activation maps contain multiple highlighted regions and the relationship between them are unclear. The activation peaks in the two views (the cyan boxes) actually correspond to different objects which leads to a failure estimation. However, the partial activation map clearly shows the relationship between these regions and provides more accurate estimation (the red line). We also present the quantitative results in Fig. 7. As can be seen from this figure, the partial activation map significantly outperforms the overall map for cross-view orientation estimation. Partial activation based orientation estimation has over 16% samples with angle error less than ±3.5° (the red bar at 0°), while overall map based method only has less than 12%.

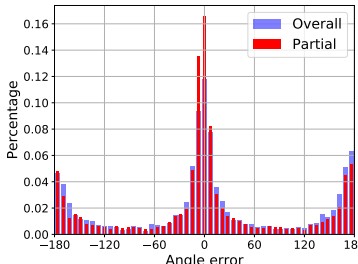

Figure 7: Estimation error distribution of overall and partial map. (Better viewed with zoom in)

## 5.4 APPLICATION II: INTERACTIVE RETRIEVAL

Verification applications like face recognition and re-ID usually retrieve images with a complete query image. However, the complete query image may be not available in some scenarios. For example, only part of a person is captured in surveillance image/video due to occlusion or viewpoint. Interactive retrieval, which retrieves images with interactively selected region of the query

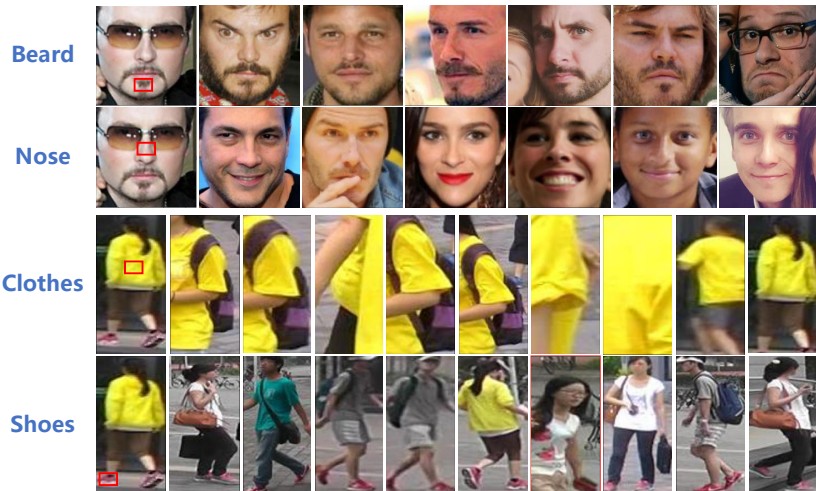

Figure 8: Top retrieved images by interactive retrieval on face and re-ID datasets. The red box on the left column indicates the query part.

image, can provide critical information for such circumstance. Since our framework provides point-to-point activation map, it can be served as a reasonable tool for measuring partial similarity. With the partial similarity, we are able to *interactively* retrieve images having similar parts with the query image instead of the overall most similar image, which may contain irrelevant or noisy information. As shown in Fig. 8, the partial activation generated by the proposed method works well on retrieving people with similar clothes and faces with similar beard. We follow the pipeline of recent approaches on face recognition (Deng et al. (2019)) and person re-identification (Luo et al. (2019)). For re-ID, the model is trained and evaluated on Market- 1501 (Zheng et al. (2015)) where some validation images only contain a small part of a person. Although the model is trained with Euclidean distance without L2 normalization, cosine similarity still works well as the evaluation metric.

For face recognition, we take the trained model on CASIA-WebFace (Yi et al. (2014)) and evaluate the model on FIW (Robinson et al. (2018)) where the face identification and kinship relationship are available. The interactive search is adopted by simply matching the equivalent partial feature in Eq. 3 ($W_{i,j}^q A_{i,j}^q$) with the reference embedding features. Fig. 8 shows that different images are retrieved when searching with different partial features on the same query image. However, we do find some failure cases as shown in the last row of Fig. 8 with the red box. There is no shoes in the failure image, but why is this image retrieved with a top rank? In the activation map of Fig. 9, three regions of the query image have a high activation on the retrieved image. And there is a high activation on the purse in the retrieved image corresponding to the shoes in the query image. This might be because of their similar red color and the arm of the retrieved person may appears like a leg from a specific viewpoint. This example also validates the importance of partial activation map generated by our framework for explanation.

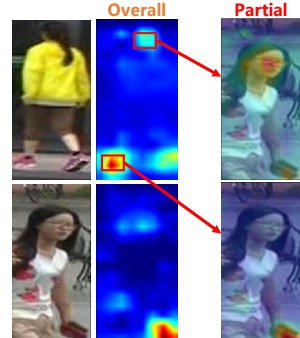

Figure 9: Explanation for the failure case by decomposition.

## 6 CONCLUSION

We propose a simple yet effective framework for visual explanation of deep metric learning based on the idea of activation decomposition. The framework is applicable to a host of applications, e.g. image retrieval, face recognition, person re-ID, geo-localization, etc. Experiments show the importance of visual explanation for metric learning as well as the superiority of both the overall and partial activation map generated by the proposed method. Furthermore, we introduce two potential applications, i.e. cross-view pattern discovery and interactive retrieval, which highlight the importance of partial activation map.

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

# A   APPENDIX

## A.1   WEAKLY SUPERVISED LOCALIZATION DETAILS

As discussed in Appendix A.4, the activation map of Grad-CAM on GMP architecture is more scattering, as larger threshold is needed for Grad-CAM to generate comparable result in our experiment.

Table 4: Localization accuracy (IOU 0.5) with different thresholds on CUB validation set.

| Threshold | Grad-CAM | Grad-CAM(no norm) | Decomposition+Bias | Decomposition |
|-----------|----------|-------------------|--------------------|---------------|
| 0.15 | **16.71**% | 16.83% | 23.49% | 17.38% |
| 0.2 | 16.26% | 17.06% | 32.01% | 19.77% |
| 0.3 | 14.48% | 21.59% | **44.94**% | 34.92% |
| 0.4 | 9.43% | 35.01% | 37.85% | **50.64**% |
| 0.5 | 4.43% | 47.00% | 21.99% | 45.78% |
| 0.6 | 1.54% | **48.27**% | 9.35% | 27.98% |
| 0.7 | 0.27% | 20.90% | 2.25% | 9.11% |

## A.2   MODEL DIAGNOSIS DETAILS

Table 5: Localization accuracy (IOU 0.5) on CUB training set with MS (multiple similarity) loss.

| Threshold | Decomposition+Bias | Decomposition |
|-----------|--------------------|---------------|
| 0.2 | 41.05% | 26.16% |
| 0.3 | **52.10**% | 44.41% |
| 0.4 | 42.77% | **58.99**% |
| 0.5 | 24.97% | 51.17% |
| 0.6 | 9.35% | 27.98% |

Table 6: Localization accuracy (IOU 0.5) on CUB training set with Triplet loss.

| Threshold | Decomposition+Bias | Decomposition |
|-----------|--------------------|---------------|
| 0.2 | 36.60% | 24.00% |
| 0.3 | **39.93**% | 38.57% |
| 0.4 | 29.03% | **55.60**% |
| 0.5 | 14.22% | 53.49% |
| 0.6 | 4.93% | 32.49% |

## A.3   TRANSFORMATION MATRIX FOR GAP AND GMP

For the feature $A \in \mathbb{R}^{m \times n \times p}$, the GAP (global average pooling) is equivalent to the flattened feature $\hat{A} \in \mathbb{R}^{mnp}$ followed by transformation matrix $T_{GAP} \in \mathbb{R}^{p \times mnp}$:

$$GAP(A) = \frac{1}{mn} \sum_{i,j} A_{i,j} = T_{GAP} \hat{A} \tag{6}$$

Here $(i, j)$ denotes the spatial coordinates. By reshaping the $T_{GAP}$ to $p \times m \times n \times p$ as $T_{GAP}^*$, the matrix is given by:

$$T_{GAP}^*(c, i, j, k) = \begin{cases} \dfrac{1}{mn} & c = k \\ 0 & c \neq k \end{cases} \tag{7}$$

The matrix $T_{GAP}$ is simply calculated by reshaping $T_{GAP}^*$ to $p \times mnp$. As shown in Section 4.2, GMP is equivalent to GAP with a maximum mask in validation phase. The transformation matrix of GMP (global maximum pooling) is given by:

$$T_{GMP}^* = mn(T_{GAP}^* \odot M) \tag{8}$$

Here $M \in R^{m \times n \times p}$ is the maximum matrix of $A$ where only the maximum position of each channel has nonzero value as 1. $T_{GMP}$ is computed by reshaping $T_{GMP}^*$ to $p \times mnp$.

## A.4 GRAD-CAM ON GMP

Considering a classification architecture with GMP and FC ($W \in \mathbb{R}^{p \times l}$ and no bias), the prediction score for class c is given by $S_c = \sum_k max_{i,j}(A_{:,:,k})W_{k,c} = \sum_{i,j,k} A_{i,j,k} M_{i,j,k} W_{k,c}$, where M is the maximum matrix in Appendix A.3. The decomposition of position $(i, j)$ is $\sum_k A_{i,j,k} M_{i,j,k} W_{k,c}$ and only the maximum position of each channel has a nonzero value. However, for Grad-CAM, the activation map is given by:

$$GradCAM_{i,j} = A_{i,j} \cdot GAP(\frac{\partial S_c}{\partial A_{i,j}}) = \frac{1}{mn} \sum_k A_{i,j,k} W_{k,c}, \tag{9}$$

where all (i,j) positions have a nonzero weight resulting in a more scattering activation map.

## A.5 GRAD-CAM FOR METRIC LEARNING

For metric learning architecture like Section 4.1, the similarity is formulated as $S = \frac{E^q \cdot E^r}{|E^q||E^r|}$, where $E^q \in \mathbb{R}^l$ and $E^r \in \mathbb{R}^l$ are the embedding vector of query and retrieved image. $|x|$ denotes the L2 norm and $a \cdot b$ is the inner product of $a$ and $b$. The Grad-CAM map of query image is given by:

$$GradCAM_{i,j} = A_{i,j}^q \cdot GAP(\frac{\partial S}{\partial A^q}) = (GAP(\frac{\partial S}{\partial A^q}))^T A_{i,j}^q \tag{10}$$

With the gradient chain rule, the gradient is written as:

$$\frac{\partial S}{\partial A^q} = \frac{\partial \frac{(E^r)^T E^q}{|E^q||E^r|}}{\partial A^q} = (\frac{E^r}{|E^r|})^T \frac{\partial(E^q/|E^q|)}{\partial E^q} \frac{\partial E^q}{\partial A^q} = (\frac{E^r}{|E^r|})^T \frac{\partial(E^q/|E^q|)}{\partial E^q} W^q \tag{11}$$

By expanding $E^r$ as $\sum_{x,y} W_{x,y}^r A_{x,y}^r + B^r$ (Section 4.1) and merging Eq. 10 with Eq. 11, the Grad-CAM map is reformulated as:

$$\begin{aligned} GradCAM_{i,j} &= (GAP((\frac{E^r}{|E^r|})^T \frac{\partial(E^q/|E^q|)}{\partial E^q} W^q))^T A_{i,j}^q \\ &= \frac{1}{Z}(\sum_{i^*,j^*}(E^r)^T \frac{\partial(E^q/|E^q|)}{\partial E^q} W_{i^*,j^*}^q) A_{i,j}^q \\ &= \frac{1}{Z}(E^r)^T \frac{\partial(E^q/|E^q|)}{\partial E^q}(\sum_{i^*,j^*} W_{i^*,j^*}^q) A_{i,j}^q \\ &= \frac{1}{Z}(\sum_{x,y} W_{x,y}^r A_{x,y}^r + B^r) \cdot (\frac{\partial(E^q/|E^q|)}{\partial E^q} GAP(W^q) A_{i,j}^q) \\ &= \frac{1}{Z}(\frac{\partial(E^q/|E^q|)}{\partial E^q} GAP(W^q) A_{i,j}^q) \cdot (\sum_{x,y} W_{x,y}^r A_{x,y}^r + B^r) \end{aligned} \tag{12}$$

Here the $Z$ is the normalization term for simplicity. $\frac{\partial E/|E|}{\partial E}$ is the $l \times l$ Jacobian matrix given by:

$$\frac{\partial(E/|E|)}{\partial E} = \Big(\frac{\partial(E_i/|E|)}{\partial E_j}\Big)_{i,j} = \begin{cases} \frac{1}{|E|}(1 - \frac{E_i^2}{|E|^2}) & i = j \\ -\frac{E_i E_j}{|E|^3} & i \neq j \end{cases} \tag{13}$$

$\frac{1}{|E|}$ is the normalization term. For dominant channel $i$, the weight $(1 - \frac{E_i^2}{|E|^2})$ is small resulting in a more scattering activation map.

## A.6 QUALITATIVE RESULTS

We provide more qualitative results of using our activation decomposition framework for the two new applications, i.e., cross-view pattern discovery and interactive retrieval.

### A.6.1 CROSS-VIEW PATTERN DISCOVERY

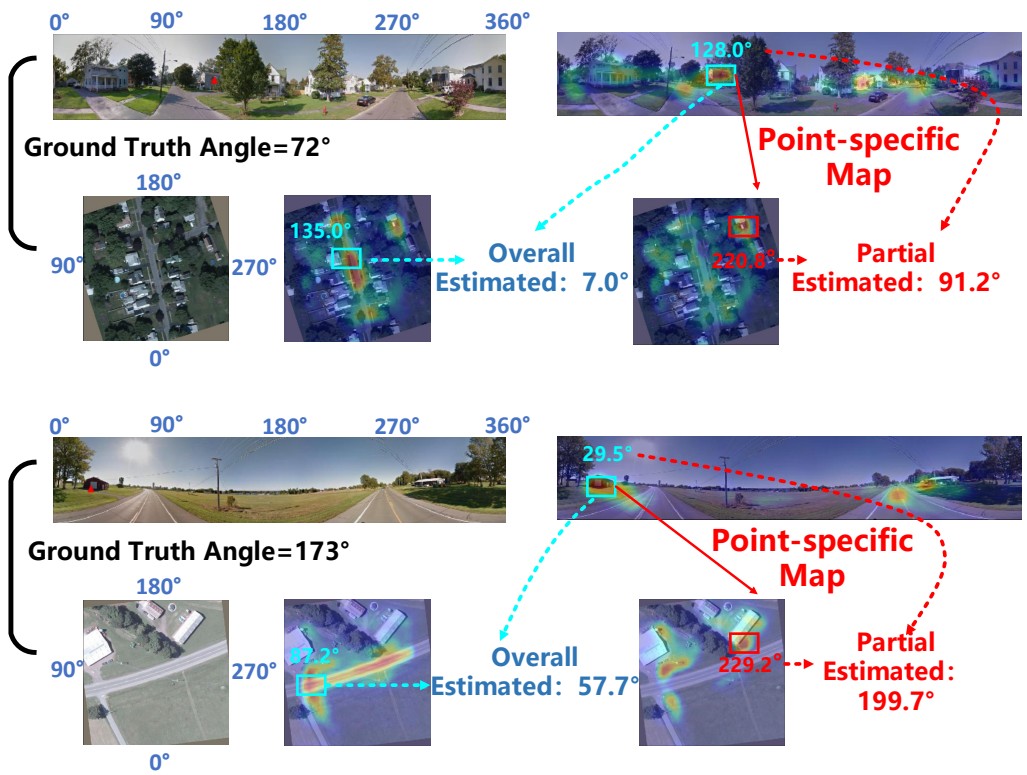

Figure 10: Examples of cross-view pattern discovery, i.e., image orientation estimation. (Best viewed in color)

### A.6.2 INTERACTIVE RETRIEVAL ON FACE

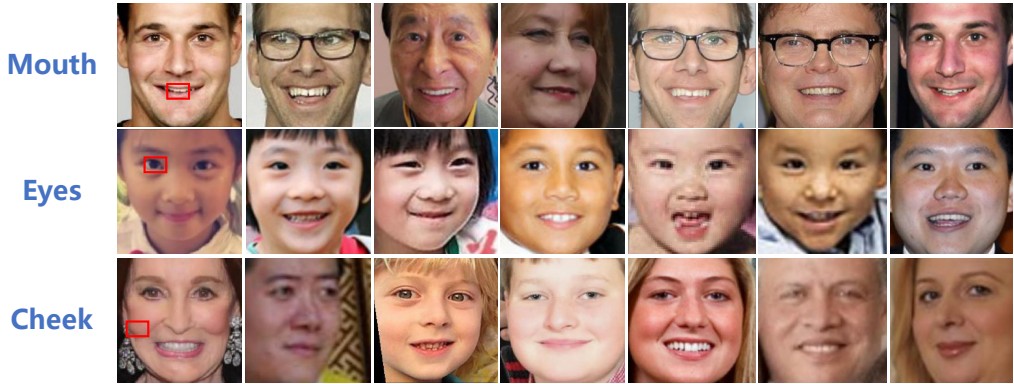

Figure 11: Top retrieved images by interactive retrieval on face recognition.

### A.6.3 INTERACTIVE RETRIEVAL ON PERSON RE-ID

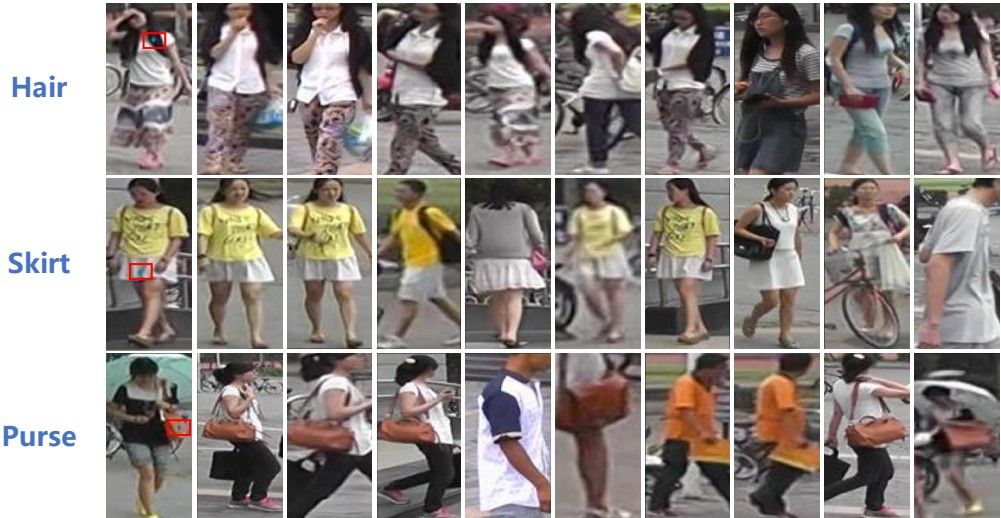

Figure 12: Top retrieved images by interactive retrieval on person re-identification.

## A.7 CROSS-VIEW PATTERN DISCOVERY DETAILS

### A.7.1 EXPLANATION OF FIGURE 6

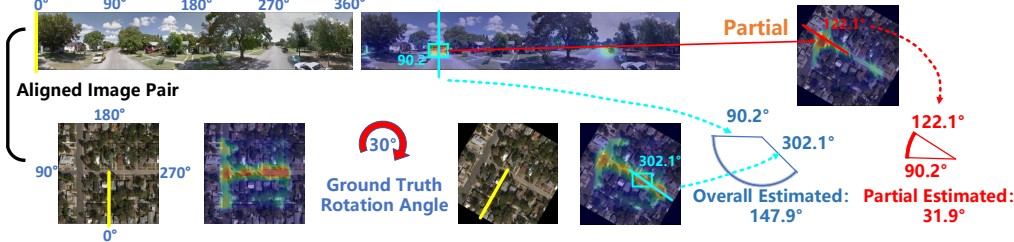

Figure 13: Example of cross-view pattern discovery, e.g. image orientation estimation. (Best viewed in color)

Here we explain Fig. 6 again (Fig. 13) with more details. We first reiterate the application and experiment setting. The dataset (CVUSA) we used in the experiment is for image-based cross-view geo-localization (Zhai et al. (2017); Hu et al. (2018)) and contains image pairs from street and aerial views. Each matched image pair corresponds to the same GPS location. The objective of geo-localization is to find the best matching GPS-tagged aerial-view image in a reference dataset for a query street-view image. In the CVUSA dataset (Zhai et al. (2017)), the image pairs are aligned in terms of orientation. For example, as shown in Fig. 13, the left two images (street-view panorama and aerial-view images) are the original image pair and the yellow line denotes the $0°$ which corresponds to the south direction ($180°$ corresponds to the North direction). We use $[0, 360]°$ to denote different angles as marked on these two images.

For the cross-view image matching/retrieval task, if the image pairs are not aligned (e.g., randomly rotate the aerial view images), we discovered that the activation map can be used for orientation estimation. Specifically, we train the Siamese image matching model with randomly rotated aerial images so that the model is rotation-invariant and so is the overall activation map for aerial image. In the example of Fig. 13, the overall activation maps for the original aligned pair are first generated to show the highlighted regions, and both views focus on the road area in this example. The most

activated regions are highly relevant in both views and most of them are similar patterns. When we randomly rotate the aerial image ($30°$ in this example), the aerial view activation map still focuses on the road area which is relevant to the street view. In this paper, we simply use the pixel with the maximum activation value for orientation estimation, because the most activated areas (highlighted in cyan boxes in Fig. 13) are likely to be relevant from two views. For the street view image, the selected pixel lies in the angle of $90.2°$ (the cyan line) based on the angle marks on the left aligned image. For the rotated aerial view, the selected pixel lies in the angle of $302.1°$ (the cyan line). Since the overall map contains multiple activated regions, the selected pixels in both views actually do not correspond to the same object, which reveals one disadvantage by only using the overall activation map. The estimated angle is $302.1° - 90.2° = 147.9°$, which is not correct. However, for the selected pixel (the one with the maximum activation value) in the street view image, if we generate its corresponding point-specific (partial) activation map on the aerial view image (as shown on the very right of Fig. 13), the new selected pixel on this point-specific activation map lies in the angle of $122.1°$ (the red line). Then, the estimated angle is calculated as $122.1° - 90.2° = 31.9°$, which is very close to the ground truth ($30°$). This demonstrates the advantage of the point-specific activation decomposition for finding more fine-grained information in this application.

### A.7.2 How to compute the angle error

The angle error is computed by $error = ground\,truth - estimated\,angle$. We then add or subtract $360°$ to this term to make it in the range of $[-180, 180]°$ if the absolute value of the error is greater than $180°$. For example, when the error is $359°$, we will subtract it by $360°$ and get $-1°$ degree as the error. This is a reasonable setting adopted from the previous work (Zhai et al. (2017)).

### A.7.3 Sensitivity of the point-specific activation map

We present a demo to show how the point-specific map changes according to the query pixel: `https://github.com/Jeff-Zilence/anonymous`. In this example, the resolution of the query image is $224 \times 1232$. As shown in the demo, the point-specific activation map changes dramatically when the query pixel moves from the left to the right in the query image because the object changes. As in this example, when the query pixel moves by 10 pixels, the object would change and the corresponding point-specific activation map on the retrieved aerial image will be quite different. By generating the point-specific activation map, we can obtain fine-grained information or interpretation for deep metric learning.

### A.8 Interactive Retrieval Details

### A.8.1 How to compute the similarity given a point of interest

Following Eq. 3, we first compute $W_{i,j}^q A_{i,j}^q$ as the feature of each position on the last convolutional layer, and a bilinear interpolation is adopted to generate the feature for every pixel of the original image. For a point of interest $(i, j)$, we compute the cosine similarity between the calculated feature on $(i, j)$ and the embedding feature of the reference images as the point-specific similarity, so the embedding features of the reference dataset do not need to be recomputed. This similarity can be also considered as the summation of the values in the point-specific map corresponding to $(i, j)$. In the case of Eq. 2 (CNN+GAP), the similarity can be simplified as $\sum_{x,y}(\sum_k A_{i,j,k}^q A_{x,y,k}^r)$.

