# OpenReview forum: "Visual Explanation for Deep Metric Learning"
_ICLR.cc/2020/Conference — Reject_

### Official Review · AnonReviewer1 · 2019-10-21
**Official Blind Review #1**

**Rating:** 3

**Review:**

This work proposes a novel approach for visualizing the predictions of neural network models on pairwise tasks, e.g. predicting whether two images are similar. The authors show that to see similarity between two images down on the pixel/region level, it is not sufficient to apply methods like Grad-CAM which do not aim for decomposition. Instead, the authors' approach namely targets decomposition, and shows the benefit of decomposition through intuitive examples and qualitative results. The authors also quantitatively show the benefit of their method. First, they measure the performance of their method vs CAM and Grad-CAM, on the weakly supervised localization (WSL) task. They also show how their method reveals the disadvantages of standard triplet loss compared to a recent metric learning loss.

My concerns:
1) While showing performance on WSL is appealing as it allows for a way to quantitatively evaluate the method, I wonder if there are other ways to evaluate, that explicitly measure the quality of the proposed technique for allowing interpretability and understanding of the base model's performance.
2) The Triplet vs MS experiment is interesting, but I'm not sure this is the most convincing way to show that this proposed visualization technique is better than something else. Just because something shows one method is worse than another, doesn't mean that this better/worse assessment is accurate. Further, how would Grad-CAM do on the same task?
3) The retrieval experiment only shows qualitative results, and again, no baseline is compared.
4) Similarity is a relative judgement; it's hard to say if two items are similar, but easier to say if A and B are more similar than A and C. It seems the proposed method doesn't consider negatives, which is perhaps a limitation.

**Experience Assessment:**

I have read many papers in this area.

**Review Assessment: Checking Correctness Of Derivations And Theory:**

I assessed the sensibility of the derivations and theory.

**Review Assessment: Checking Correctness Of Experiments:**

I assessed the sensibility of the experiments.

**Review Assessment: Thoroughness In Paper Reading:**

I read the paper thoroughly.

---

> ### Author Response · Authors · 2019-11-08
> **Response**
>
> We sincerely thank you for your appreciation and valuable suggestions.. Our response for the mentioned concerns are listed below:
>
> 1) Firstly, as the first quantitative study on visual explanation of deep metric learning and its practical applications, it is hard to find annotated dataset for evaluating the quality of generated activation map. Since the WSL is widely used in explanation methods for classification and the annotation is available, WSL can be one of the best choices for evaluation. Other evaluation like human evaluation (seem to be explicit but may be subjective) can be expensive and time-consuming. Secondly, the theoretical advantage of the proposed method over the existing method, i.e. GradCAM, has been clearly illustrated in Section 4.3, the empirical result serves as the validation evidence. Our theoretical analysis suggests that directly adopting GradCAM with L2 normalization would result in scattering activation map which is supported by the result of section 5.1 (see Table 1, Fig. 4). Even the GradCAM without L2 normalization may have issues with GMP or flattened feature based architecture as shown in Appendix A.4. We argue the result of WSL in Section 5.1 is enough to show the superiority of proposed method for overall activation map. Besides, the point-specific activation map is another advantage which is  not provided by GradCAM.
>
> 2) Firstly, as stated in the response 1, the advantage of proposed method over GradCAM has been theoretically stated in Section 4.3 and empirically supported with the result in Section 5.1. The point of Section 5.2 is to show a new way for model diagnosis instead of comparing with other methods, since the performance of different loss functions are only evaluated by the retrieval accuracy and distribution. We show that our method can provide additional valuable information about the generalization ability by only checking the activation map on the training set.  Secondly, as stated in Section 4.3, GradCAM involves a heuristic step which makes the meaning of the generated activation map not clear and can be irrelevant to the model, thus is not applicable for model diagnosis. In other words, GradCAM may generate aesthetically pleasing activation maps for a given model, while how the map is related to the model is unclear due to the heuristic nature. However, our method clearly shows the contribution of each region to the overall similarity score, which is a reasonable way to check whether the model is working well or not.
>
> 3) The similarity between different regions in images is highly subjective and hard to annotate. Note that the interactive retrieval is a new application, and it is impossible to provide quantitative results on the existing dataset and compare with other methods, since there is no straightforward baseline method and evaluation metrics. New dataset may be proposed to evaluate this task in future work. More qualitative results are included in the updated version (Appendix A.6).
>
> 4) For deep metric learning, the similarity score (or distance) measures the similarity of two images. If A and B have a higher score than A and C, then we say A is more similar to B than to C. As for face recognition and re-ID, whether two images are from the same face or the same person is very clear because of ground-truth. Although the relative similarity is an interesting topic, it is beyond the scope of this paper. This paper aims to show how much each region contributes to the overall similarity and the relative similarity may be considered in future work. By saying negatives, we guess the reviewer means the images which are not positive pairs, e.g. images from different persons. Although negative pairs have relatively low similarity score, they still have the regions which contribute the most to the similarity so that our method would still generate activation map. In fact, our method can generate the visual explanation for arbitrary image pairs including negative pairs, so we don't think the negatives can be considered as a limitation.

---

### Official Review · AnonReviewer2 · 2019-10-24
**Official Blind Review #2**

**Rating:** 8

**Review:**

This paper proposes a visualization method for deep metric learning, which derived by analyzing the inner product of two globally averaged activations. The proposed method can generate an overall activation map which highlights the regions contributing most to the similarity. Also, it can generate a partial activation map that lights the regions in one image that have significant activation responses on a specific potion in the other image. The authors also analyzed the linearly of the fully connected layers and global max pooling. These contributions make the applicability of CAM to many CNN architectures. Further, the metric learning architecture is extended to Grad-CAM map, and the problem of Grad-CAM map is pointed out. To the best of my knowledge, these contributions are novel, and derivations seem to be correct.

Experiments on weakly-supervised localization, model diagnosis, and the applications of the proposed decomposition model in cross-view pattern discovery and interactive retrieval are promising.

Overall, this paper is well written, and contributions are good.

Minor problems.
In Sec.1 and Sec.2.2, the authors wrote the Grad-CAM has been used for visualization of re-ID (Gordo & Larlus (2017)). However, this paper seems to be not the works of Grad-CAM nor re-ID.

In my understanding, Decomposition+Bias is a more accurate model than Decomposition.
In the experiments of the Sec5.1 and 5.2, the performances of Decompsotion+Bias are lower than Decomposition. However, there are no explanations for this reason.


**Experience Assessment:**

I have read many papers in this area.

**Review Assessment: Checking Correctness Of Derivations And Theory:**

I assessed the sensibility of the derivations and theory.

**Review Assessment: Checking Correctness Of Experiments:**

I assessed the sensibility of the experiments.

**Review Assessment: Thoroughness In Paper Reading:**

I read the paper at least twice and used my best judgement in assessing the paper.

---

> ### Author Response · Authors · 2019-11-08
> **Response**
>
> We sincerely thank you for your appreciation and valuable suggestions. Our response for the mentioned concerns are listed below:
>
> 1) Thank you for pointing out this mistake, Gordo & Larlus (2017) utilized GradCAM for visualizing image retrieval result (not re-ID). We have corrected this point in the manuscript.
>
> 2) We use the same pre-trained model with two different settings for generating activation map, i.e. "Decomposition" (setting 1) and "Decomposition+Bias" (setting 2), so we didn't use a more accurate model for "Decomposition+Bias". The model is the same. As mentioned in Section 4.1, the bias term is often ignored in existing works,, so we study the empirical results of both settings. The "Decomposition+Bias" setting contains more terms, but the additional information may not be helpful, since more information does not guarantee more accurate result. As in the weakly supervised localization experiments, the performance of "Decomposition+Bias" is not as good as "Decomposition", which means the additional information is not helpful for this task. A possible explanation is that the bias $B$ is the same for any input images, which means the bias term actually provides prior knowledge about the image distribution, since different embedding features may have different similarities with the bias $B$. The prior knowledge is learned from the retrieval task, and it may not work for weakly supervised localization.

---

### Official Review · AnonReviewer3 · 2019-10-25
**Official Blind Review #3**

**Rating:** 6

**Review:**

= Summary
This paper presents a simple method that draws visual attention of deep embedding networks for metric learning. It basically follows the class attention mapping strategy based on global pooling operation [Zhou et al., CVPR 2016], but extends the original version to point-specific attention which is novel and enable interesting applications on image retrieval. In addition, the proposed method seems also independent of the loss function used for metric learning, thus it can be applied to most of existing deep embedding networks to understand their behaviors in a qualitative manner.


= Decision
My current decision is officially "weak reject" but "borderline" in my mind. The major concern of mine is its weaknesses in clarity and technical novelty. However, I still believe this submission is valuable since it addresses a relatively new and timely topic, the proposed method is simple yet effective, and the applications of point-specific attention (i.e., "cross-view pattern discovery" and "interactive retrieval") are all interesting and practically useful. If the clarity issues are all clearly addressed, I would upgrade my rating.


= Comments
[Pros]
1) The motivation and implementation of the point-specific attention are convincing.
2) The point-specific attention enables not only image-to-image retrieval, but also more elaborate understanding about a pair of images and their similarity.
3) The proposed technique is model-agnostic and loss-agnostic, thus can be applied to most of existing deep embedding networks for image retrieval. Also, the proposed technique does not degrade the retrieval performance.
4) The proposed technique is simple yet effective in multiple applications, most of which are practically useful and have great impact. For example, the weakly supervised localization is an essential step towards many weakly supervised approaches for higher-level recognition tasks like semantic segmentation, and the interactive retrieval will allow us to build more realistic and useful image retrieval systems.

[Cons]
1) The proposed technique itself is not new but a straightforward extension of an existing work [Zhou et al., CVPR 2016].
2) The manuscript is not crystal clear.
- The name of the proposed point-specific attention (i.e., "Partial attention") is misleading.
- The way to compute the point-specific attention map is not clearly described in Section 3.
- It is hard to understand the contents in Figure 6 as they are not clearly illustrated.
- The experimental and implementation details of the last two applications are not given. For example: (cross-view pattern discovery) how to compute the angle error, and how much the proposed technique is sensitive to the position selected on the query image, (interactive retrieval) how to compute the similarity between images given a specific region of interest on query.
3) More qualitative results on the last two applications should be presented, even in an appendix, to convince future readers. Especially, in the case of "interactive retrieval", more results are demanded as quantitative performance analysis seems not straightforward.


= Post-rebuttal review
The rebuttal resolves my major concerns and the manuscript has been carefully revised accordingly. So as I promised in my original review, I upgrade the score to weak accept.

**Experience Assessment:**

I have published one or two papers in this area.

**Review Assessment: Checking Correctness Of Derivations And Theory:**

I assessed the sensibility of the derivations and theory.

**Review Assessment: Checking Correctness Of Experiments:**

I carefully checked the experiments.

**Review Assessment: Thoroughness In Paper Reading:**

I made a quick assessment of this paper.

---

> ### Author Response · Authors · 2019-11-08
> **Response part 1**
>
> We sincerely thank you for your appreciation and valuable suggestions. Our response for the mentioned concerns are listed below:
>
> 1) The original CAM is widely considered as a visual explanation for classification based on a specific architecture, that is “CNN + GAP (global average pooling) + FC”. When computing the activation map for standard VGG with global max pooling (GMP) and three fc layers, [Zhou et al., CVPR 2016] has to replace these components with GAP+FC and retrain the model, which is not required for our method. In fact, the technique of CAM cannot be straightforwardly applied to other architectures, while our idea of activation decomposition can. By considering CAM as a special case of activation decomposition on GAP+FC based architectures, we propose a unified framework for activation decomposition on architectures beyond GAP+FC, and the proposed framework is not limited to classification problem. On metric learning and its applications, the proposed activation decomposition shows a unique advantage (supplying point-specific activation map) over the overall activation map. In Section 3, we use a simple architecture (CNN+GAP) to better illustrate the pipeline of our framework, but our method is NOT limited to this particular architecture. Section 4 clearly shows how to compute the activation map for architectures beyond GAP+FC, e.g. flattened feature+FC for face recognition and GMP+FC for image retrieval in this paper. In other words, our method applies to many typical CNN architectures which have covered most of the SOTA metric learning architectures. Since the more complicated architectures considered in this paper are beyond the capacity of CAM, we don't think the proposed method is a straightforward extension of CAM.
>
> 2) -About the "partial" term: We never use the term “partial attention” in this paper, we only use “partial activation map”  and “point-specific map”. In contrast to the overall activation map, the point-specific activation map is sometimes referred to as "partial activation map" in this paper (i.e., interchangeable). The meaning of this term has been defined in Section 1, but it may be misleading to some readers. Here, we try to explain it more clearly. For query image A and retrieved image B, the similarity between A and B is denoted as S. Our idea is to first decompose this S along A and B so that the regions which contribute the most to S are highlighted in both A and B. We call this map “the overall activation map” in this paper, see the second column (Overall) of Fig. 1 for an example. Then for a point of interest (i,j) in A, we further decompose the value at (i,j) on the overall map of A along image B, so that the regions in image B which contribute the most to the overall map value at (i,j) of image A are highlighted, see the third column (Partial) of Fig. 1. We call this map the “partial (or point-specific) activation map” of point (i,j) in this paper.
>
> - About the way to compute point-specific map: As mentioned in Section 3, for each query point $(i,j)$ in the query image, the corresponding point-specific activation map in the retrieved image is given by $\sum_{k}A^{q}_{i,j,k}A^{r}_{x,y,k}$, while more explanation can be helpful for understanding this term. For each point $(i,j)$ in the query image, the point-specific map consists of the activation intensity between the $(i,j)$ in the query image and each $(x,y)$ in the retrieved image. The value of activation map at position $(x,y)$ is given by $\sum_{k}A^{q}_{i,j,k}A^{r}_{x,y,k}$ and the bilinear interpolation is implemented to generate the full resolution activation map. We would add more explanation in the manuscript if the reviewers think it's necessary.
>
> - About the angle error: The angle error is computed by "error = ground truth - estimated angle". We then add or subtract 360 to this term to make it in the range of [-180,180] if the absolute value of the error is greater than 180. For example, when the error is 359 degrees, we will subtract it by 360 and get -1 degree as the error. This is a reasonable setting adopted from the previous work (Zhai et al. (2017)). We have added this setting in Appendix (A.7.2).
>
> (continued in part 2)

---

> > ### Author Response · Authors · 2019-11-08
> > **Response part 2**
> >
> > - About the Figure 6: More explanation about Figure 6 has been added in the Appendix (A.7.1) as requested.  Here, we first reiterate the application and experiment setting. The dataset (CVUSA) we used in the experiment is for image-based cross-view geo-localization (Zhai et al. (2017); Hu et al. (2018)) and contains image pairs from street and aerial views. Each matched image pair corresponds to the same GPS location. The objective of geo-localization is to find the best matching GPS-tagged aerial-view image in a reference dataset for a query street-view image. In the CVUSA dataset  (Zhai et al. (2017)), the image pairs are aligned in terms of orientation. For example, as shown in Fig. 6, the left two images (street-view panorama and aerial-view images) are the original image pair and the yellow line denotes $0^{\circ}$ which corresponds to the South direction ($180^{\circ}$ corresponds to the North direction). We use $[0,360]^{\circ}$ to denote different angles as marked on those two images.
> >
> > For the cross-view image matching/retrieval task, if the image pairs are not aligned (e.g., randomly rotate the aerial view images), we discovered that the activation map can be used for orientation estimation. Specifically, we train the Siamese image matching model (Hu et al. (2018)) with randomly rotated aerial images so that the model is rotation-invariant and so is the overall activation map for aerial image. In the example of Fig. 6, the overall activation maps for the original aligned pair are first generated to show the highlighted regions, and both views focus on the road area in this example. The most activated regions are highly relevant in both views and most of them are similar patterns. When we randomly rotate the aerial image ($30^{\circ}$ in this example), the aerial view activation map still focuses on the road area which is relevant to the street view. In this paper, we simply use the pixel with the maximum activation value for orientation estimation, because the most activated areas (highlighted in cyan boxes in Fig. 6) are likely to be relevant from two views. For the street view image, the selected pixel lies in the angle of $90.2^{\circ}$ (the cyan line) based on the angle marks on the left aligned image. For the rotated aerial view, the selected pixel lies in the angle of $302.1^{\circ}$ (the cyan line). Since the overall map contains multiple activated regions, the selected pixels in both views actually do not correspond to the same object, which reveals one disadvantage of using only the overall activation map. The estimated angle is $302.1^{\circ}-90.2^{\circ} = 147.9^{\circ}$, which is not correct. However, for the selected pixel (the one with the maximum activation value) in the street view image, if we generate its corresponding point-specific (partial) activation map on the aerial view image (as shown on the very right of Fig. 6), the new selected pixel on this point-specific activation map lies in the angle of $122.1^{\circ}$ (the red line). Then, the estimated angle is calculated as $122.1^{\circ}-90.2^{\circ}=31.9^{\circ}$, which is very close to the ground truth ($30^{\circ}$). This demonstrates the advantage of the point-specific activation decomposition for finding more fine-grained information in this application.
> >
> > - About the sensitivity of pixel selection: In the cross-view pattern discovery, we select the pixel with the maximum activation value because the most activated regions are likely to be relevant on two views. We didn't find other straightforward selection strategy, so only the maximum pixel selection was evaluated. If we randomly select pixels in both views, the overall map won't be able to generate meaningful orientation estimation result.
> > Hope this answers your question. However, we are not completely clear about what you mean by “sensitivity of pixel selection”. Could you make it more clear?
> >
> > - About the similarity between images and a specific region of interest on query: Following Eq.3, we first compute ($W^{q}_{i,j}A^{q}_{i,j}$) as the feature of each position on the last convolutional layer, and a bilinear interpolation is adopted to generate the feature for every pixel of the original image. For a point of interest (i,j), we compute  the cosine similarity between the calculated feature on (i,j) and the embedding feature of the reference images as the point-specific similarity, so the embedding features of the reference dataset do not need to be recomputed. This similarity can be also considered as the summation of the values in the point-specific map corresponding to (i,j). In the case of Eq.2 (CNN+GAP), the similarity can be simplified as $\sum_{x,y}(\sum_{k}A^{q}_{i,j,k} A^{r}_{x,y,k})$. We have added these details in Appendix (A.8.1).
> >
> > 3) More qualitative results have been included in the Appendix (A.6) as requested.

---

### Author Response · Authors · 2019-11-15
**Summary of update**

Since we haven’t received any reply from the reviewers in the discussion phase, we summarize our updates as follows:

1) As requested by reviewer 3, we add more qualitative results of cross-view pattern discovery and interactive retrieval in Appendix (A 6).
2) We add more explanation and details about cross-view pattern discovery and interactive retrieval in Appendix (A 7 and A 8).
3) We correct the mistake pointed out by reviewer 2 in Section 1 and 2.2.
4) We are not sure if we understand the concern of reviewer 3 about the sensitivity correctly and request for clarification. Since we haven’t received any response from the reviewer, we tried our best to explain it. We also add two animated GIF images at  https://github.com/Jeff-Zilence/anonymous to demonstrate the sensitivity of the point-specific map along the query pixel. Explanation is included in Appendix (A 7.3).  Hope this answers the question of reviewer 3.

For reviewer 3: We would like to emphasize that our method is Not limited to the GAP architecture like CAM. For better illustration, we show how to compute the activation map on GAP in Section 3, but we also clearly show our pipeline on other architectures beyond GAP in Section 4. Hope this clarifies the misunderstanding about our method.

---

### Decision · Program_Chairs · 2019-12-19

**Decision:**

Reject

**Comment:**

This submission proposes a method for providing visual explanations for why two images match by highlighting image regions that most contribute to similarity.

Reviewers agreed that the problem is interesting but were divided on the degree of novelty of the proposed approach.

AC shares R1’s concern that localization accuracy is not satisfactory as a quantitative measure of the quality of the explanations. In particular, it pre-supposes what the explanations ought to be, i.e. that a good explanation means good localization. A small user-study would be more convincing. A more convincing evaluation would also include a study of explanation of image pairs with different degrees of similarity (e.g. images that are dissimilar as well as images with the same object).

AC also shares R2’s concern about the validity of the model diagnosis application. This discussion also relies on the assumption that better localization of the whole object means a better explanation. Further, the highlighted regions in Figure 5 are very similar. Once again, a user study would help to indicate whether these results really do improve explainability.

Reviewers also had concerns about missing details and, while the authors did improve this, key details are still missing. For example, the localization method that was used was only referenced but should be described in the paper itself.

Given that several concerns remain, AC recommends rejection.